# Osteochondral Regeneration Ability of Uncultured Bone Marrow Mononuclear Cells and Platelet-Rich Fibrin Scaffold

**DOI:** 10.3390/bioengineering10060661

**Published:** 2023-05-31

**Authors:** Tung Nguyen-Thanh, Bao-Song Nguyen-Tran, Sara Cruciani, Thuy-Duong Nguyen-Thi, Thuan Dang-Cong, Margherita Maioli

**Affiliations:** 1Faculty of Basic Science, Hue University of Medicine and Pharmacy, 6 Ngo Quyen Street, Hue 49000, Vietnam; 2Institute of Biomedicine, Hue University of Medicine and Pharmacy, 6 Ngo Quyen Street, Hue 49000, Vietnam; 3Department of Histology, Embryology, Pathology and Forensic, Hue University of Medicine and Pharmacy, 6 Ngo Quyen Street, Hue 49000, Vietnam; ntbsong@huemed-univ.edu.vn (B.-S.N.-T.); dcthuan@huemed-univ.edu.vn (T.D.-C.); 4Department of Biomedical Sciences, University of Sassari, Viale San Pietro 43/B, 07100 Sassari, Italy; sara.cruciani@outlook.com; 5Faculty of Odonto-Stomatology, Hue University of Medicine and Pharmacy, 6 Ngo Quyen Street, Hue 49000, Vietnam; nttduong@huemed-univ.edu.vn

**Keywords:** osteochondral regeneration, bone marrow mononuclear cells, platelet-rich fibrin, tissue regeneration, osteogenesis

## Abstract

Objectives: Platelet-rich fibrin (PRF) and bone marrow mononuclear cells are potential scaffolds and cell sources for osteochondral regeneration. The main aim of this paper is to examine the effects of PRF scaffolds and autologous uncultured bone marrow mononuclear cells on osteochondral regeneration in rabbit knees. Materials and Methods: Three different types of PRF scaffolds were generated from peripheral blood (Ch-PRF and L-PRF) and bone marrow combined with uncultured bone marrow mononuclear cells (BMM-PRF). The histological characteristics of these scaffolds were assessed via hematoxylin–eosin staining, PicroSirius red staining, and immunohistochemical staining. Osteochondral defects with a diameter of 3 mm and depth of 3 mm were created on the trochlear groove of the rabbit’s femur. Different PRF scaffolds were then applied to treat the defects. A group of rabbits with induced osteochondral defects that were not treated with any scaffold was used as a control. Osteochondral tissue regeneration was assessed after 2, 4, and 6 weeks by macroscopy (using the Internal Cartilage Repair Society score, X-ray) and microscopy (hematoxylin—eosin stain, safranin O stain, toluidine stain, and Wakitani histological scale, immunohistochemistry), in addition to gene expression analysis of osteochondral markers. Results: Ch-PRF had a heterogeneous fibrin network structure and cellular population; L-PRF and BMM-PRF had a homogeneous structure with a uniform distribution of the fibrin network. Ch-PRF and L-PRF contained a population of CD45-positive leukocytes embedded in the fibrin network, while mononuclear cells in the BMM-PRF scaffold were positive for the pluripotent stem cell-specific antibody Oct-4. In comparison to the untreated group, the rabbits that were given the autologous graft displayed significantly improved healing of the articular cartilage tissue and of the subchondral bone. Regeneration was gradually observed after 2, 4, and 6 weeks of PRF scaffold treatment, which was particularly evident in the BMM-PRF group. Conclusions: The combination of biomaterials with autologous platelet-rich fibrin and uncultured bone marrow mononuclear cells promoted osteochondral regeneration in a rabbit model more than platelet-rich fibrin material alone. Our results indicate that autologous platelet-rich fibrin scaffolds combined with uncultured bone marrow mononuclear cells applied in healing osteochondral lesions may represent a suitable treatment in addition to stem cell and biomaterial therapy.

## 1. Introduction

Cartilage regeneration continues to pose difficulties in tissue engineering because it does not contain neural and vascular components and thus displays limited restoration after injury [1,2,3]. Regenerative medicine has opened wide perspectives for the treatment of cartilage damage, involving three essential integrated components: cell sources, scaffolds, and growth factors [4,5,6,7].

Autologous chondrocytes have been widely applied in the regeneration of articular cartilage, even though they have some disadvantages such as the lack of cell sources for large lesions and the risk of dedifferentiation during in vitro culture [8]. Mesenchymal stem cells (MSCs) are regarded as a superior alternative cell source and have shown encouraging results in the field of articular cartilage tissue regeneration owing to their high capacity for proliferation and differentiation as well as to their simple separation processes from a variety of mesenchymal tissues [9,10]. However, the expansion of MSCs requires high facilities and technical expertise that might not be accessible in many locations. Recently, autologous bone marrow mononuclear cells (BMMCs) have been used as promising cell therapies for various diseases in a wide spectrum of medications. Previous research in animal as well as clinical trials has indicated the efficacy and safety of BMMCs in the treatment of cartilage lesions, which if compared to MSCs, show the advantages of easy isolation and no requirement of in vitro expansion before application [11,12,13,14].

Platelet-rich fibrin (PRF) scaffolds are second-generation platelet concentrates [15]. They contain different growth factors that contribute to the regeneration of soft and hard tissue, stored in granules in the platelets, including platelet-derived growth factor (PDGF), insulin-like growth factor 1 (IGF-1), transforming growth factor β (TGF-β), epidermal growth factor (EGF) [16] and vascular endothelial growth factor (VEGF) [17]. PRF is also considered a fibrin biomaterial. Because of its molecular structure and low concentration of thrombin, it serves as a suitable matrix for the migration of endothelial cells and fibroblasts [18]. Furthermore, the autologous origin and on-hand accessibility of PRF could lead to the practicality of this material and can minimize the duration of surgical operations. Numerous authors have concentrated their attention on the application of PRF in the tissue engineering of cartilage and tendons, which has resulted in an increase in in vitro, preclinical, and clinical research [14,19,20]. Chondrogenesis depends on several cartilage-specific markers, including collagen, aggrecan (ACAN), SRY-box transcription factor 9 (SOX9), and osteogenic-related markers, such as alkaline phosphatase (ALPL), bone gamma-carboxyglutamate protein (BGLAP), and RUNX family transcription factor 2 (RUNX2), which participate in tissue regeneration [21,22,23,24,25].

Little information is available on the combination of PRF scaffolds and autologous uncultured BMMCs in healing cartilage lesions. Therefore, we performed this research to examine the effects of PRF scaffolds and autologous uncultured BMMCs on osteochondral regeneration in rabbit knees.

## 2. Methods

### 2.1. Animals

Thirty-eight New Zealand white rabbits (weight: 2–2.5 kg) were used in this study. The use of rabbits in the present study was authorized by The Hue University Animal Ethics Committee (Certificate Reference number: HU VN0010, 10 November 2021). They were divided at random into four categories: nine rabbits were used for experiments in which osteochondral defects were treated with Choukroun’s platelet-rich fibrin (Ch-PRF) scaffold, nine rabbits were used for experiments in which osteochondral defects were treated with leukocyte platelet-rich fibrin (L-PRF) scaffold, nine rabbits were used for experiments in which osteochondral defects were treated with scaffolds combining uncultured bone marrow mononuclear cells and platelet-rich fibrin from bone marrow (BMM-PRF), and nine rabbits were used as controls in which osteochondral defects were created and any scaffold treatment was applied.

### 2.2. Preparation of Fibrin Scaffold and Bone Marrow Mononuclear Cells

#### 2.2.1. Preparation of Choukroun’s Platelet-Rich Fibrin Scaffold from Peripheral Blood

According to Choukroun’s method with modifications, 9 mL blood from the rabbit’s central ear artery was collected and then immediately moved into a 15 mL polypropylene centrifuge tube without anticoagulant. Blood samples were centrifuged at 2700 rpm (400× *g*) for 12 min. After centrifugation, whole blood in each tube was separated into three layers: platelet-poor plasma (PPP), platelet-rich fibrin (PRF), and red blood cells at the base. The plasma and red blood cells were discarded, the Ch-PRF scaffold was placed in a sterile, perforated metal mesh, and a light metal plate was placed on the gel to make a consistent thickness of 1 mm.

#### 2.2.2. Preparation of Leukocyte Platelet-Rich Fibrin Scaffold from Peripheral Blood

According to O’Cornell, with modifications, 9 mL blood from the rabbit’s central ear artery was collected and then immediately transferred into EDTA blood collection tubes. These samples were then moved to a 15 mL polypropylene centrifuge tube and centrifuged at 2000 rpm for five minutes. Following the first centrifugation, the sample was separated into 3 distinct layers: an upper layer containing plasma, platelets, and white blood cells; a middle layer containing mainly white blood cells; and a bottom layer (hematocrit) consisting mainly of red blood cells. The upper and middle layers were switched to a new tube and centrifuged at 4000 rpm for 5 min. After this second centrifugation step, the samples were separated into plasma and white cell pellets of leukocytes: the upper plasma layer, known as platelet-poor plasma (PPP), and the bottom layer, known as platelet-rich plasma (PRP). The PPP layer was discarded. Leukocytes were resuspended in platelet-rich plasma (PRP). Subsequently, calcium chloride (10%) was given to obtain a final concentration of 0.1% in plasma. This produced an L-PRF matrix. The L-PRF scaffold was put in a sterile perforated metal mesh, and a light metal plate was placed on the gel to ensure a consistent thickness of 1 mm.

#### 2.2.3. Preparation of Scaffolds Combining Uncultured Bone Marrow Mononuclear Cells and Platelets-Rich Fibrin from Bone Marrow

According to Soh Nishimoto’s method with modifications, rabbits were anesthetized by intramuscular injection of ketamine (35.0 mg/kg) and xylazine (5.0 mg/kg). Bone marrow was taken from the posterior aspect of two iliac crests of the rabbits by using an 18 G biopsy needle attached to a 10 mL syringe, pre-loaded with 1 mL citrate and 2.0 mL Dulbecco’s phosphate-buffered saline (DPBS). Three milliliters of bone marrow from each iliac crest was collected and immediately mixed with the solution (3 mL) in a syringe by gentle rotation. A total of 12 mL of bone marrow solution from two iliac crests was equally transferred into two 15 mL conical tubes containing 3 mL Ficoll–Paque medium (density 1.077 g/mL) in each tube. These tubes were centrifuged at 2000 rpm for 20 min. Following the first centrifugation, the bone marrow sample was separated into four distinct layers: a top layer containing plasma, a buffy coat containing mostly bone marrow mononuclear cells, a Ficoll–Paque fraction, and a bottom layer consisting mostly of granulocytes and erythrocytes. The plasma and buffy coat layers were taken from two tubes and put into a new 15 mL conical tube. This tube was centrifuged at 2000 rpm for 20 min. After the second centrifugation step, the samples were separated into plasma and pellets of BMMCs; the upper plasma layer is known as bone marrow platelet-poor plasma (PPP), and the bottom layer is known as bone marrow platelet-rich plasma (PRP). The bone marrow platelet-poor plasma layer (PPP) was removed, and the BMMC pellet was resuspended in PRP. Subsequently, calcium chloride (10%) was given to obtain a final concentration of 0.1% in plasma. This produced a BMM-PRF matrix. The BMM-PRF gel was placed in a sterile perforated metal mesh, and a light metal plate was placed on the gel to ensure a consistent thickness of 1 mm.

#### 2.2.4. PicroSirius Red Staining for Assessing the Fibrin Structure of the PRF Scaffold

Fibrin architecture was detected by PicroSirius red staining as previously reported [26]. Briefly, the PRF slides were deparaffinized, hydrated, treated with adequate picrosirius red solution (Abcam, ab246832), and incubated for 60 min. Finally, the stained slides were mounted on a resinous medium.

#### 2.2.5. Immunohistochemical Staining for Detecting Cells Distributing in PRF Scaffold

Immunohistochemistry was performed as previously described [27]. Heat-induced epitope retrieval was performed in a retrieval buffer (Dako Target Retrieval 10×, S1699, Dako, USA), and endogenous peroxidase was blocked using a peroxidase-blocking solution (Dako REAL Peroxidase-Blocking Solution, S2023, Dako, USA). Next, the PRF sections were incubated overnight with primary antibodies against fibrinogen (1:100, Santa Cruz, sc-69775), CD45, and Oct-4 (1:100, Santa Cruz, sc-1178), and then incubated for 45 min with biotinylated secondary antibodies (1:1000). Visualization was achieved using 3-amino-9-ethyl, a high-sensitive substrate chromogen (AEC+ High-Sensitivity Substrate Chromogen, ready-to-use, K346911-2, DAKO, Glostrup, Denmark).

### 2.3. Gene Expression Analysis

Total RNA was extracted from each sample and quantified using a NanoDrop One/OneC Microvolume UV–Vis spectrophotometer (Thermo Fisher Scientific, Grand Island, NY, USA). Approximately 1 µg of total RNA was reverse-transcribed using a high-capacity cDNA reverse transcription kit (Thermo Fisher Scientific, Grand Island, NY, USA). Real-time quantitative PCR was performed in triplicate using a CFX Thermal Cycler (Bio-Rad, Hercules, CA, USA). Amplification cycling was set as specified in the KAPA SYBR^®^ FAST protocol: 95 °C for 3 min, followed by cycling at 95 °C for 3 s, and 60 °C for a total of 40 cycles. The target Ct values of each sample were normalized to r18S, which was used as the reference gene. The relative values of the genes of interest were expressed as the fold change (2^−∆∆Ct^) of mRNA levels observed in normal tissue. The primers used were obtained from Thermo Fisher Scientific (Grand Island, NY, USA) and are listed in Table 1.

### 2.4. Surgical Procedures for Inducing Rabbit’s Knee Osteochondral Defects

Intramuscular administration of 40 mg/kg ketamine hydrochloride with 5 mg/kg xylazine hydrochloride was performed to induce anesthesia. In the knee joint of each rabbit’s right posterior limb, a medial peripatellar incision was performed, and the patella was separated laterally under sterile conditions. Next, an osteochondral defect on the trochlear groove of the femur was generated by using a biopsy punch to determine the location and the size, and then, a dental stainless drill was utilized to create the defect with a size of 3 × 3 mm (diameter × depth). The bleeding in defects was identified, indicating subchondral bone involvement. The surgical procedures as well as the defect size were similar to those described previously [28,29].

Ch-PRF, L-PRF, and BMM-PRF scaffolds were then inserted into the defects of three different rabbit groups (n = 9 for each group). Within the control group, there were nine rabbits that were not implanted with scaffolds. The wounds were stitched up in three different layers. To prevent wound infections, gentamicin, at a dose of 4 mg/kg, was administered through subcutaneous injection to the rabbits for a period of three days following surgical procedures. The rabbits were cared for in separate cages, monitored, and evaluated daily to determine the overall status as well as the condition of the incision.

At 2-, 4- and 6-weeks post-surgery, rabbits were euthanized, and the femur’s distal parts with defects were obtained. All samples were macroscopically and histologically assessed.

### 2.5. Macroscopic Assessment

At the time of collection, the samples were photographed and assessed by the Internal Cartilage Repair Society score (ICRS) macroscopic assessment score for the cartilage regeneration [30]. This scale integrates three characteristics of the healing of osteochondral defects on macroscopy: the degree of defect repair, integration to the border zone, and macroscopic appearance. The overall repair evaluation was scored and graded from 1 to 12 points as follows: grade 1, normal 12; grade 2, nearly normal 11-8, Grade 3: abnormal 7-4, and grade 4, severely abnormal.

The distal parts of the femurs were also radiologically evaluated using an EZ Dent X-ray machine. The X-ray sensor was positioned near the lateral edge of defects and then captured using a 65 kVp, 7.5 mA current for 0.25 s. Biomedical software EZ Dent was utilized for analyzing X-ray results.

### 2.6. Microscopic Assessment

All samples were fixated in 10% neutral buffered formalin solution for 48 h before being placed in 10% EDTA solution (Sigma-Aldrich, Darmstadt, Germany) for 8 weeks to decalcify bone tissue. After that, the samples went through tissue processing and were cut into 4-micrometer slices. These slices were then stained with hematoxylin-eosin (Sigma-Aldrich, Darmstadt, Germany), 0.3% toluidine blue (Sig-ma-Aldrich, Darmstadt, Germany), and 0.25% Safranin O (Sigma-Aldrich, Darmstadt, Germany) for cartilage regeneration assessment on microscopy. The regeneration of cartilage was histologically assessed using the Wakitani scale including cell morphology, matrix staining, surface regularity, cartilage thickness, and integration of the donor cartilage with the host adjacent cartilage [9].

### 2.7. Statistical Analysis

ICRS and Wakitani scores were analyzed using Excel (Microsoft Office 365, WA, USA) and SPSS software V22.0 (SPSS Inc., Chicago, IL, USA) to identify the effect of biomaterial scaffolds on cartilage regeneration in rabbits. For each parameter, differences between the experimental and control groups were tested for significance using the Mann–Whitney *U* test; a level of significance was established at 5%. The GraphPad Prism program (version 9.0, GraphPad, San Diego, CA, USA), was utilized for the purpose of gene expression analysis. Kruskal–Wallis rank sum and one-way analysis of variance ANOVA tests with Tukey’s correction were utilized; the level of significance was established at 5%.

## 3. Results

### 3.1. Histological Characteristics of PRF Scaffold from Peripheral Blood and Bone Marrow Aspirate

We assessed the histological structure of three types of PRF scaffolds from peripheral blood and bone marrow via hematoxylin–eosin staining, Sirius red staining, and immunohistochemistry staining with a fibrinogen marker. The results showed that Ch-PRF had a heterogeneous fibrin network structure and cellular population (Figure 1a,d,g). The L-PRF scaffold had a uniform fibrin structure, with a moderate distribution of cells embedded in the fibrin network (Figure 1b,e,h). In Ch-PRF and L-PRF, a proportion of leukocytes was embedded in the fibrin structure, as demonstrated by immunohistochemistry with a CD45 marker. The distribution of leukocytes in the L-PRF scaffold was more evident than that in the Ch-PRF (Figure 1j,k).

The BMM-PRF scaffold generated from bone marrow aspirate had a homogeneous structure with a uniform distribution of the fibrin network, with BMMCs and a few red blood cells embedded in the fibrin structure (Figure 1c,f,i). Bone marrow mononuclear cells embedded in the BMM-PRF scaffold were positively stained with the pluripotent stem cell-specific antibody, Oct-4, as shown in Figure 1l.

### 3.2. Gene Expression Analysis of Osteochondral Repair

The expression levels of the main osteochondral markers were analyzed using qPCR after 2, 4, and 6 weeks. SOX9 (Figure 2a) increased from the first 2 weeks in the BMM-PRF group, while for the Ch-PRF and L-PRF groups, its expression increased only after 6 weeks, as compared to controls. Moreover, SOX9 directly regulates the expression of type II collagen (COL2A1), whose levels showed the same trend, being upregulated starting from 2 weeks in the BMM-PRF group and the longer period for Ch-PRF and L-PRF groups, as compared to controls (Figure 2g). ALPL was slightly increased in all samples analyzed, reaching a significant upregulation only in the L-PRF group after 6 weeks. A completely different pattern was seen for ACAN (Figure 2c) and COL1A1 (Figure 2f), whose expression increased in the first 2 weeks and reached a plateau state between 4 and 6 weeks for all samples analyzed. However, the highest levels of these markers were observed for the BMM-PRF group, as compared to controls. Finally, BGLAP (Figure 2d) and RUNX2 (Figure 2e) showed more constant levels over time, being slightly upregulated as compared to controls, for all treatments and observation times.

### 3.3. Macroscopic Assessment of Osteochondral Regeneration

The macroscopic assessment of osteochondral regeneration is demonstrated in Figure 3. In the untreated rabbits, after 2 weeks, the defect’s bases were covered by white-pinkish tissue with partially narrower in diameter. Four weeks post-surgery, the sizes of defects were smaller compared with the two-week cohort. After 6 weeks, the renewal tissue was seen to fill about half of the defects (Figure 3a–c). In the Ch-PRF and L-PRF groups, the repair process of osteochondral defects was relatively similar. However, there was no distinct difference in the macroscopic appearance between the defects in these two groups and the control group during the study period (Figure 3d–i). At all times of observation, the BMM-PRF group demonstrated a greater degree of reconstruction of the defects than the other groups. Compared to the other groups, this group’s defects were shallower, and their surfaces were less rough. The regeneration results of osteochondral defects were most evident in this group 6 weeks after surgery, showing that the defects were nearly covered by repaired tissue and had relatively flat surfaces (Figure 3j–l).

The macroscopic features of the control group and experimental groups were analyzed according to the ICRS macroscopic evaluation scale, as depicted in Figure 4. Two weeks after surgery, the ICRS scores of the untreated group, Ch-PRF group, L-PRF group, and BMM-PRF group were 2.3 ± 0.6, 2.7 ± 0.6, 2.7 ± 0.6, and 4.7 ± 0.6, respectively. The BMM-PRF group had a significantly higher ICRS score than the other two groups. With Ch-PRF and L-PRF rabbits, their ICRS scores did not differ significantly from the score of the control group. At 4 weeks after surgery, the ICRS scores of the untreated group, Ch-PRF group, L-PRF group, and BMM-PRF group were 3.7 ± 0.6, 4.7 ± 0.6, 5.3 ± 0.6, and 5.7 ± 0.6, respectively. The BMM-PRF and L-PRF groups scored significantly higher on the ICRS compared to the control group, whereas the score of the Ch-PRF group did not significantly outperform the control group. The difference in ICRS scores between the BMM-PRF, Ch-PRF, and L-PRF groups was not statistically significant. After 6 weeks, the ICRS scores of the untreated group, Ch-PRF group, L-PRF group, and BMM-PRF group were 5.3 ± 0.6, 6.3 ± 0.6, 6.7 ± 0.6, and 8.7 ± 0.6, respectively. The BMM-PRF group had a substantially higher ICRS score compared to the other three groups. However, the Ch-PRF group and L-PRF group did not differ significantly from the control group in the ICRS score. The assessment of subchondral bone formation in osteochondral defects by radiography is shown in Figure 5. After 2 weeks, the bottom boundary of the defects was not well-defined, indicating a peripheral bone remodeling reaction. This phenomenon was most obvious in the BMM-PRF models (Figure 5a–j). Four and six weeks post-surgery, the size of bone defects gradually narrowed, and the empty spaces were replaced by trabecular bones. This result was more noticeable in the experimental groups compared to the untreated rabbits and was most evident in the BMM-PRF group (Figure 5b–k).

### 3.4. Microscopic Evaluation of the Osteochondral Regeneration

The histology of osteochondral defects at 2-, 4-, and 6-weeks following surgery was analyzed by hematoxylin–eosin staining, as shown in Figure 6 (2 weeks), Figure 7 (4 weeks), and Figure 8 (6 weeks). After 2 weeks, in the untreated rabbits, the defects were covered by fibrous connective tissue, and there was a relatively high concentration of fibroblast cells interspersed throughout the matrix (Figure 6a–c). In the experimental groups, the grafted materials were almost completely resorbed. The repaired tissue was fibrous connective tissue, and there was no clear difference between these groups and the untreated group (Figure 6d–l). After 4 weeks post-surgery, the fibrous connective tissue became less dense in the control group, and the layer underneath demonstrated the growth of adipose niches, chondrocytes, and the cartilage matrix. Subchondral bone was absent in this group (Figure 7a–c). The experimental groups exhibited a similar occurrence. However, the creation of hyaline cartilage tissue (chondrocytes and cartilage matrix) was more apparent within the BMM-PFR group compared to the untreated group. In addition, subchondral bone appeared in these groups, with discontinuous trabeculae in the Ch-PRF and L-PRF groups, and was more evident in the BMM-PRF group with continuous trabeculae (Figure 7d–l). Six weeks post-surgery, the remaining fibrous connective tissue in the control group was considerably thinner compared to the four-week group. Cartilage tissue as well as ossified bone appeared as slightly more noticeable, although they were dispersed and inconsistent. Discontinuous trabeculae of the subchondral bone underneath could be seen (Figure 8a–c). In the experimental groups, the defects were covered by cartilage tissue more clearly than those of the untreated rabbits. In the Ch-PRF and L-PRF groups, premature chondrocytes were mostly located in the lower half of the renewal tissue (Figure 8d–i), whereas the regenerated tissue in the BMM-PRF group nearly resembled hyaline cartilage, with premature chondrocytes dispersed from the bottom to the upper layer of the renewal tissue (Figure 8j–l). In addition, the subchondral bone at 6 weeks post-surgery in these groups was easily observed, with continuous trabeculae. Especially in the BMM-PRF group, the subchondral bone trabeculae were continuous, and the intertrabecular spaces were smaller, which nearly resembled the normal tissue.

Safranin O staining results are depicted in Figure 9. Safranin O binds to cartilaginous proteoglycans and exhibits an orange-yellow color. Toluidine blue binds to cartilaginous proteoglycans and is blue in color. The toluidine blue staining results are shown in Figure 10. In the untreated rabbits, safranin O and toluidine blue were negative in the defects at 2 weeks postoperatively. They were relatively mildly positive and sporadic at 4 and 6 weeks after surgery (Figure 9b–d and Figure 10b–d), whereas safranin O and toluidine blue staining were quite similar among Ch-PRF and L-PRF groups. The result was negative at 2 weeks postoperatively, slightly positive at 4 weeks postoperatively, and clear after 6 weeks. After 6 weeks, the regeneration tissue in the L-PRF group stained with safranin O or toluidine blue was more apparent than that in the Ch-PRF group. The staining of safranin O and toluidine blue after 6 weeks in these two groups was more obvious than that in the untreated rabbits (Figure 9e–j and Figure 10e–j). As regards BMM-PRF rabbits, the staining of safranin O and toluidine blue two weeks postoperatively was like that in the other groups. However, after 4 weeks and especially after 6 weeks, safranin O and toluidine blue were positive in the renewed tissue, and the staining of safranin O as well as toluidine blue at 6 weeks postoperatively was comparable to the appearance found in natural articular cartilage (Figure 9k–m and Figure 10k–m).

Histological regeneration of osteochondral defects was analyzed using the Wakitani scale, as shown in Figure 11. At 2 weeks after surgery, the histology scores of the untreated group, Ch-PRF group, L-PRF group, and BMM-PRF group were 11.3 ± 1.2, 10.3 ± 0.6, 10.3 ± 0.6, and 9.3 ± 0.6, respectively. Nonetheless, neither the experimental groups nor the experimental and control groups differed significantly from one another. Four weeks after surgery, the histology scores of the untreated group, Ch-PRF group, L-PRF group, and BMM-PRF group were 8.3 ± 0.6, 6.3 ± 0.6, 6.3 ± 0.6, and 4.7 ± 0.6, respectively. Compared to the untreated rabbits, the histology scores of the three experimental groups were substantially lower, indicating improved osteochondral defect restoration. There was no significant difference between the Ch-PRF group and the L-PRF group, whereas the BMM-PRF group had a significantly lower histology score compared to the other two groups. Six weeks following surgery, histology scores of the untreated group, Ch-PRF group, L-PRF group, and BMM-PRF group were 6.7 ± 0.6, 5.3 ± 0.6, 5.3 ± 0.6, and 3.3 ± 0.6, respectively. The L-PRF and Ch-PRF groups exhibited no statistically significant differences regarding the Wakitani score. The BMM-PRF group had a significantly lower histology score than the control, Ch-PRF, and L-PRF groups.

## 4. Discussion

Biocompatibility is the key to the successful application of tissue-engineered tissues. Our study successfully demonstrated the biocompatibility and safety of autologous PRF scaffolds and uncultured BMMC as grafted biomaterials for cartilage repair in vivo. This is due to the autologous origin of these materials. During the experimental period, adjacent articular cartilage did not show any degenerative changes. In addition, in the experimental groups, neither an inflammatory reaction nor giant cells (frequently observed in the inflammatory response to foreign bodies) were observed in the grafted area. Two weeks postoperatively, the grafted materials were almost completely resorbed. Our results showed that rabbits treated with the PRF scaffold combined with BMMCs had a more significant effect on osteochondral defect regeneration than the control and PRF alone. This was demonstrated both macroscopically and microscopically.

PRF scaffolds sorely have been demonstrated to positively affect osteochondral healing in terms of macroscopic and histological outcomes. Many studies have been conducted on rabbits and other animal models [19,20,31,32]. PRF’s therapeutic effect is primarily attributable to its high concentration of platelet-derived protein molecules that are primarily stored and released by platelet α-granules. Among these, platelet-derived growth factors, including platelet-derived GF (PDGF), transforming GF-β1 (TGF-β1), and insulin-like GF (IGF-1), contribute significantly to the formation of cartilage tissue [33,34,35]. By modulating cell growth, neo-angiogenesis, inflammation, and deposition of extracellular matrix (ECM), they can serve as potent promoters of chondrogenesis and tendons [36,37,38]. A dense polymerized fibrin network formed within the PRF allows for increased entrapment of circulating cytokines and growth factors. These molecules are slowly released, providing long-term effects at the injury site [17]. Moreover, this release is facilitated via the creation of new growth factors from PRF membrane-dwelling leukocytes. Among the different types of leukocytes, lymphocytes are more concentrated than others. Lymphocytes are local regulators during the healing process, which might clarify the reason that PRF membranes can continue to generate significant amounts of growth factors over a long period [39].

BMMCs have been shown to have therapeutic benefits on tissue regeneration. Many in vivo studies and clinical trials of BMMC transplantation have been performed for several degenerative disorders, such as orthopedic or traumatic disorders, heart diseases, and bone lesions [40,41,42]. In the field of cartilage regeneration, Bekkers demonstrated that osteochondral defect treatment in goat models with BMMCs and chondrocytes promotes greater regeneration of the defects on macroscopy than microfractures [43]. In 2014, Song et al. compared the effect of bone marrow mesenchymal stem cells (BMSCs) and BMMCs on osteoarthritis treatment using a sheep model. The results showed that BMSCs generated higher-quality cartilage repair than BMMC. However, they suggested that BMMC is a suitable alternative to BMSCs in osteoarthritis treatment [44]. In a study by Mohamed Salem in 2020, the healing of osteochondral defects in a group treated with BMMCs combined with PRF was significantly superior to that of other groups (PRF, BMMC alone, and control groups) in both macroscopic and microscopic examinations [14]. BMMC comprises a diverse population of cells, such as mesenchymal stem cells (MSCs), hematopoietic progenitor cells (HPCs), hematopoietic stem cells, and other cells. Among these, a very small proportion (0.01%–0.001%) of the total mononuclear cells represents the mesenchymal stem cells [45]. Therefore, the efficacy of BMMCs in tissue regeneration may not be completely attributable to their MSCs component. Non-MSC elements could exert a significant influence in this process [44]. The hypothesis is further contributed through the findings of a study by Joel K. Wise, in which fresh uncultured BMMCs exhibited the same level of osteogenesis as MSCs that had been cultured and expanded when encapsulated in three-dimensional (3D) collagen–chitosan microbeads [40]. In addition, compared to BMSCs, which require a cell culture process with many high-quality requirements and which faces various other risks, BMMCs can be administered directly without requiring in vitro culture. This can reduce treatment time and expenses, in addition to the decrease in differentiation and migratory capacity caused by in vitro culture, as well as the risk of contamination and other undetermined issues.

## 5. Conclusions

In conclusion, the combination of autologous platelet-rich fibrin and uncultured bone marrow mononuclear cells promotes osteochondral regeneration within rabbit models, rather than platelet-rich fibrin material alone. Using autologous platelet-rich fibrin scaffolds combined with uncultured bone marrow mononuclear cells in healing osteochondral lesions might represent a valuable approach in addition to stem cell and biomaterial therapy.

## Figures and Tables

**Figure 1 bioengineering-10-00661-f001:**
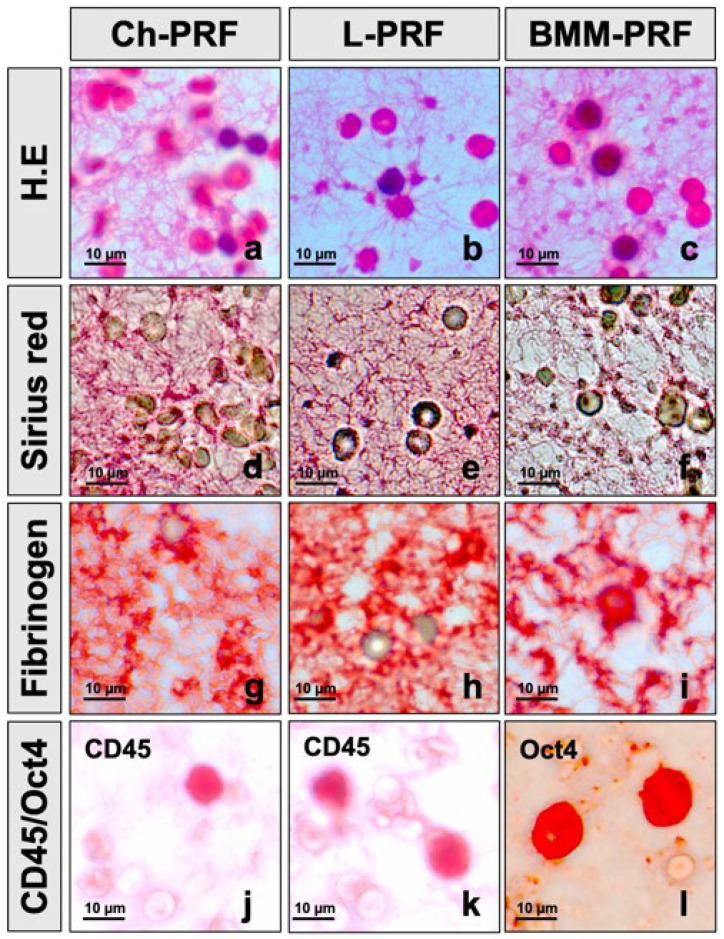
Histological structure and cell components of PRF biomaterial from peripheral blood and bone marrow aspirate. (**a**–**c**) Hematoxylin–eosin staining of Ch-PRF, L-PRF, and BMM-PRF; (**d**–**f**) Sirius red staining of Ch-PRF, L-PRF, and BMM-PRF; (**g**–**i**) immunohistochemistry staining of Ch-PRF, L-PRF, and BMM-PRF with fibrinogen; (**j**–**l**) immunohistochemistry staining of Ch-PRF, L-PRF, and BMM-PRF with CD45 and Oct-4.

**Figure 2 bioengineering-10-00661-f002:**
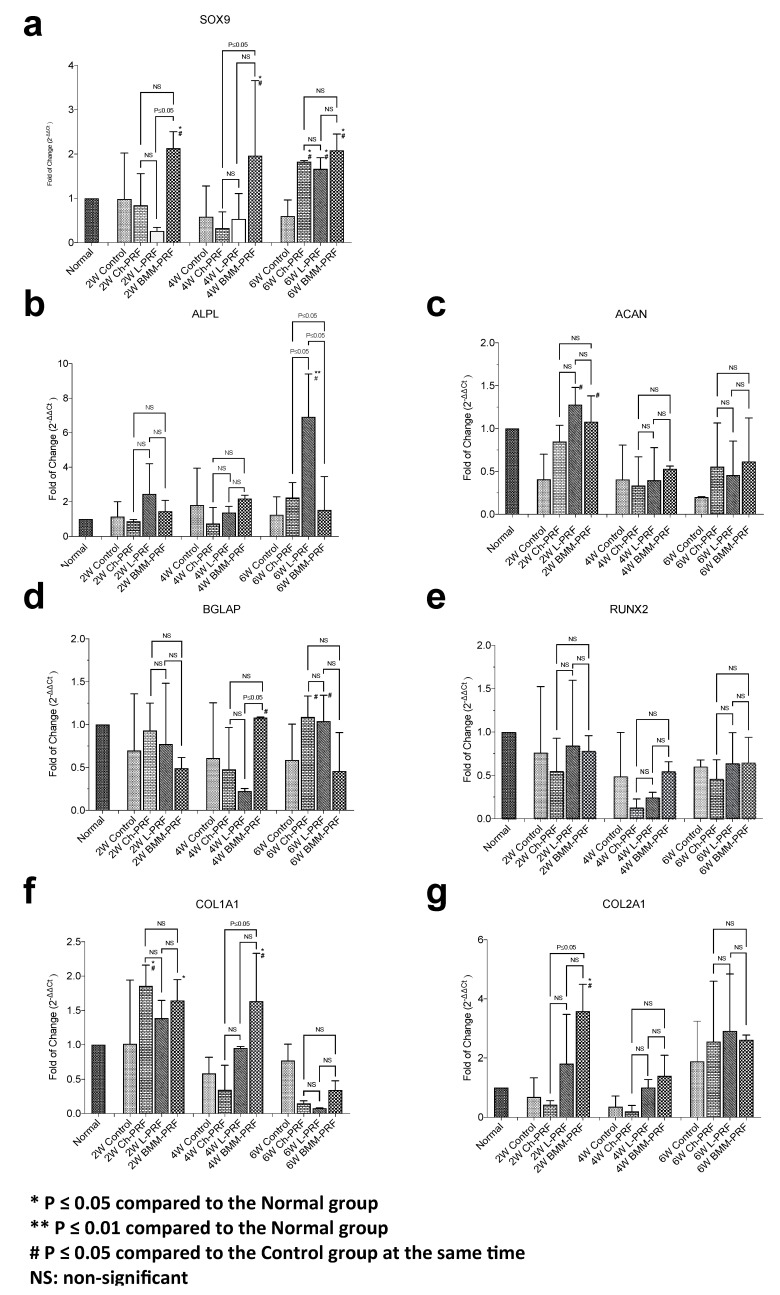
Expression of osteochondral markers. The expression of SOX9 (**a**), ALPL (**b**), ACAN (**c**), BGLAP (**d**), RUNX2 (**e**), COL1A1 (**f**), and COL2A1 (**g**) was evaluated after 2, 4, and 6 weeks postoperatively in the control group, Ch-PRF group, L-PRF group, and BMM-PRF group. The mRNA levels for each gene were normalized to 18S and expressed as a fold of change (2^−∆∆Ct^) of the mRNA levels observed in normal samples defined as 1. Data are expressed as mean ± SD referred to the control. (* *p* ≤ 0.05 compared to Normal group; ** *p* ≤ 0.01 compared to Normal group; # *p* ≤ 0.05 compared to Control group at the same time).

**Figure 3 bioengineering-10-00661-f003:**
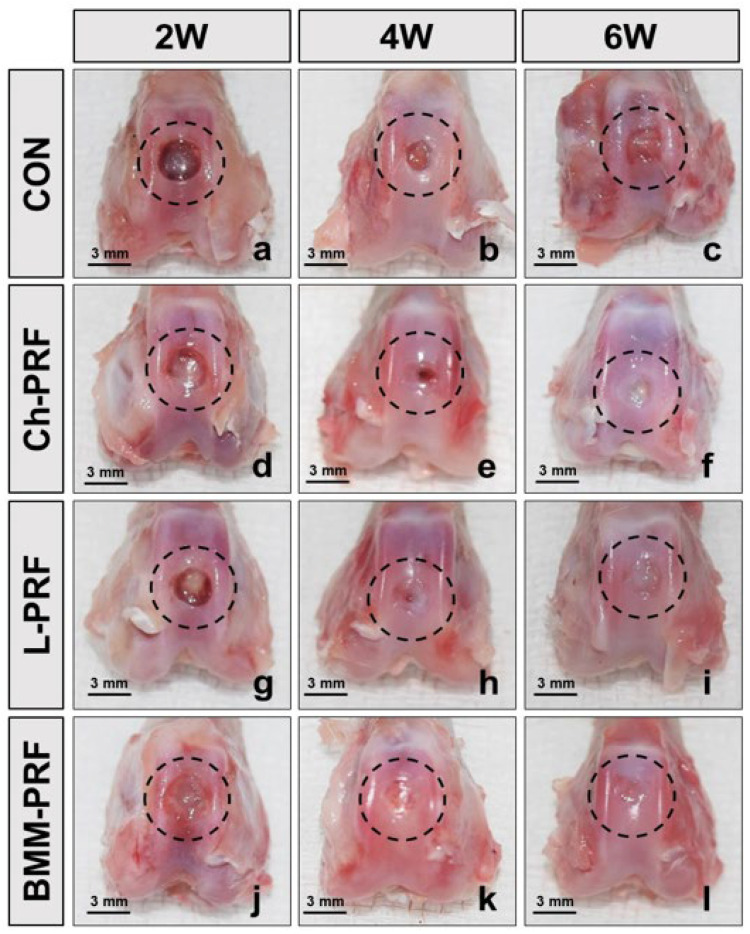
Macroscopic appearance of osteochondral defects repairs after 2, 4, and 6 weeks postoperatively in the control group (**a**–**c**), Ch-PRF group (**d**–**f**), L-PRF group (**g**–**i**), and BMM-PRF group (**j**–**l**). (The dotted circle represents the defect area).

**Figure 4 bioengineering-10-00661-f004:**
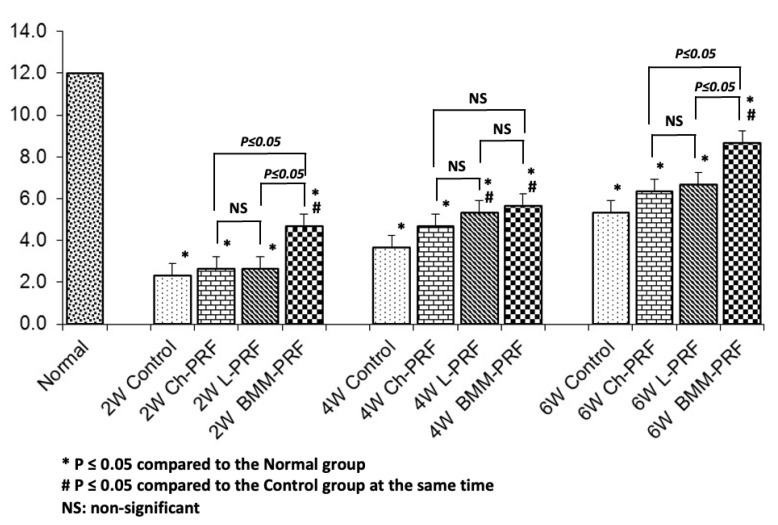
ICRS scale-based macroscopic evaluation of the regeneration of osteochondral defects at 2, 4, and 6 weeks after surgery. (* *p* ≤ 0.05 compared to Normal group; # *p* ≤ 0.05 compared to Control group at the same time).

**Figure 5 bioengineering-10-00661-f005:**
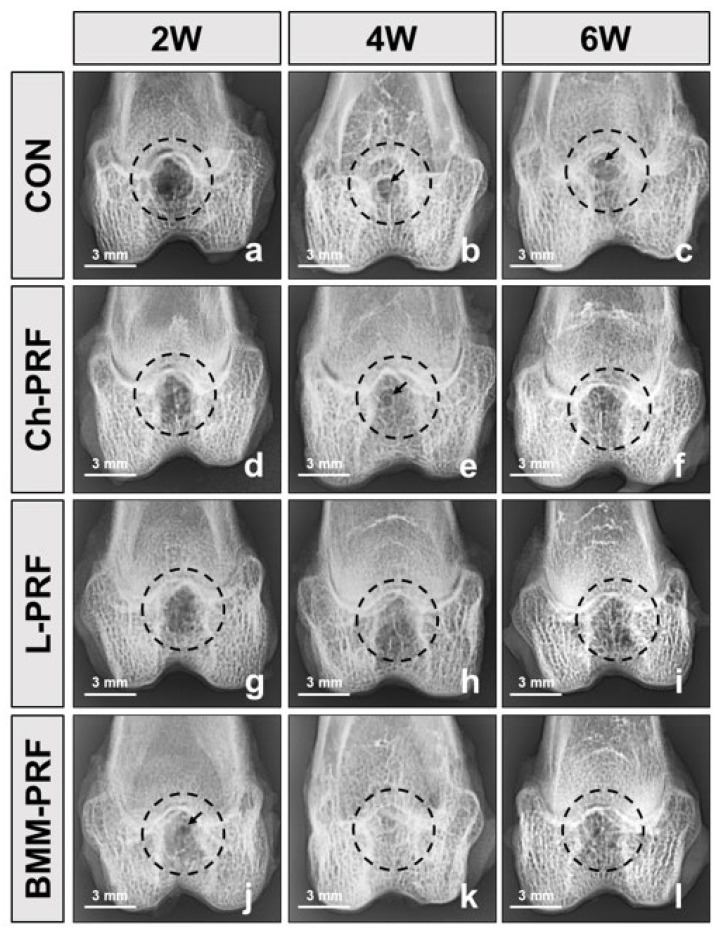
The anteroposterior radiologic finding of the subchondral bone regeneration at 2, 4, and 6 weeks after surgery in the control group (**a**–**c**), Ch-PRF group (**d**–**f**), L-PRF group (**g**–**i**), and BMM-PRF group (**j**–**l**). (The dotted circle represents the defect area).

**Figure 6 bioengineering-10-00661-f006:**
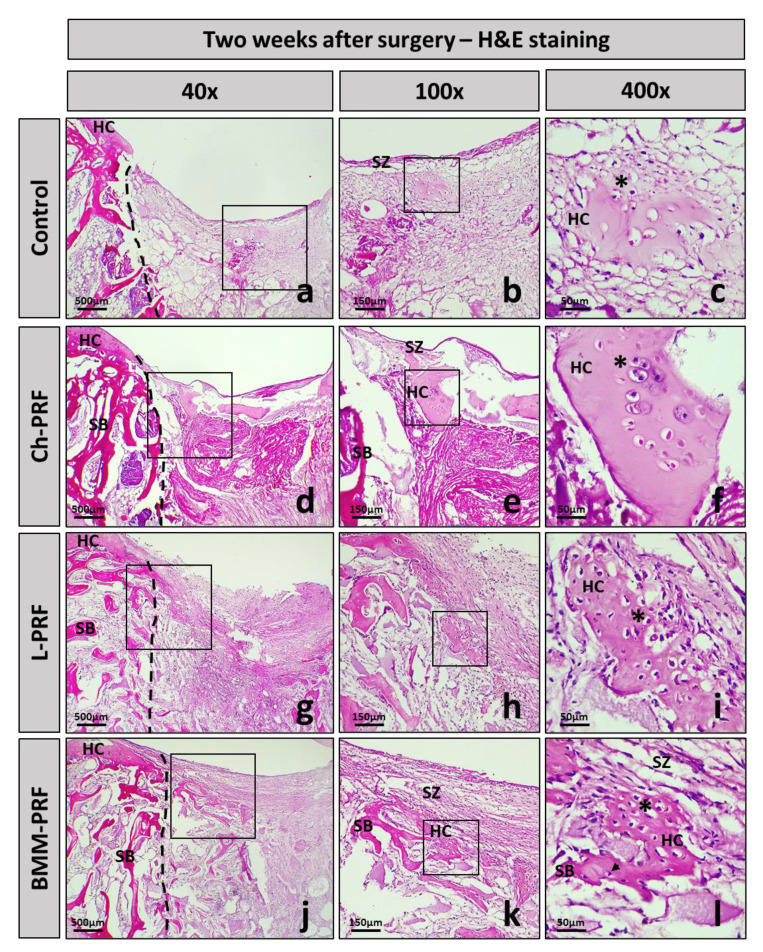
Histological manifestation of osteochondral defects after 2 weeks. Images of H&E staining of the control group (**a**–**c**), Ch-PRF group (**d**–**f**), L-PRF group (**g**–**i**), and BMM-PRF group (**j**–**l**) at 2 weeks following surgery (at 40, 100, and 400 magnification). The box indicates the amplifier area; **HC**, hyaline cartilage; **SZ**, superficial zone; **SB**, subchondral bone; chondrocytes (asterisk); osteocytes (arrowhead).

**Figure 7 bioengineering-10-00661-f007:**
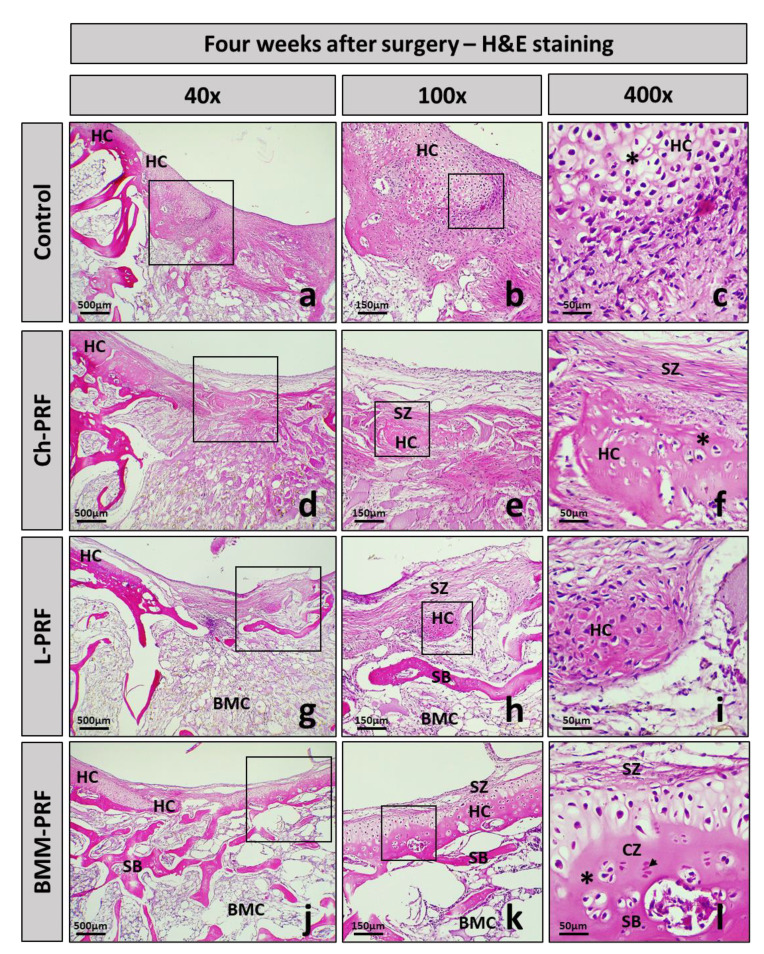
Histological manifestation of osteochondral defects after 4 weeks. Images of H&E staining of the control (**a**–**c**), Ch-PRF (**d**–**f**), L-PRF (**g**–**i**), and BMM-PRF groups (**j**–**l**) at 4 weeks following surgery (at 40, 100, and 400 magnification). The box indicates the amplifier area; **HC**, hyaline cartilage; **SZ**, superficial zone; **DZ**, deep zone; **CZ**, calcified zone; **SB**, subchondral bone; **BMC**, bone marrow cavity; chondrocytes (asterisk); osteocytes (arrowhead).

**Figure 8 bioengineering-10-00661-f008:**
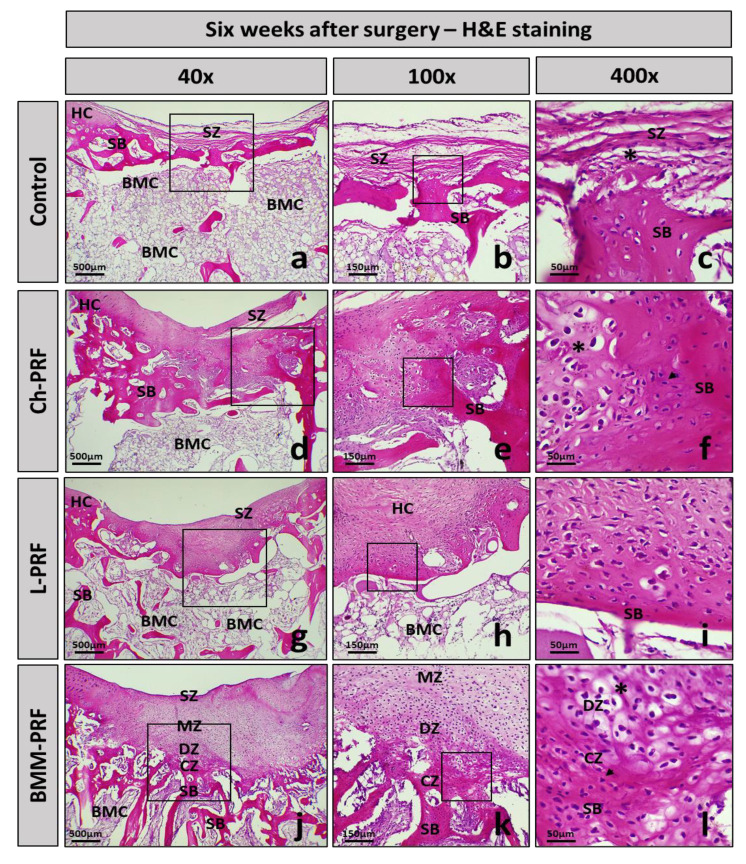
Histological manifestation of osteochondral defects after 6 weeks. Images of H&E staining of the control (**a**–**c**), Ch-PRF (**d**–**f**), L-PRF (**g**–**i**), and BMM-PRF groups (**j**–**l**) at 6 weeks following surgery (at 40, 100, and 400 magnification). The box indicates the amplifier area; **HC**, hyaline cartilage; **SZ**, superficial zone; **DZ**, deep zone; **CZ**, calcified zone; **SB**, subchondral bone; **BMC**, bone marrow cavity; chondrocytes (asterisk); osteocytes (arrowhead).

**Figure 9 bioengineering-10-00661-f009:**
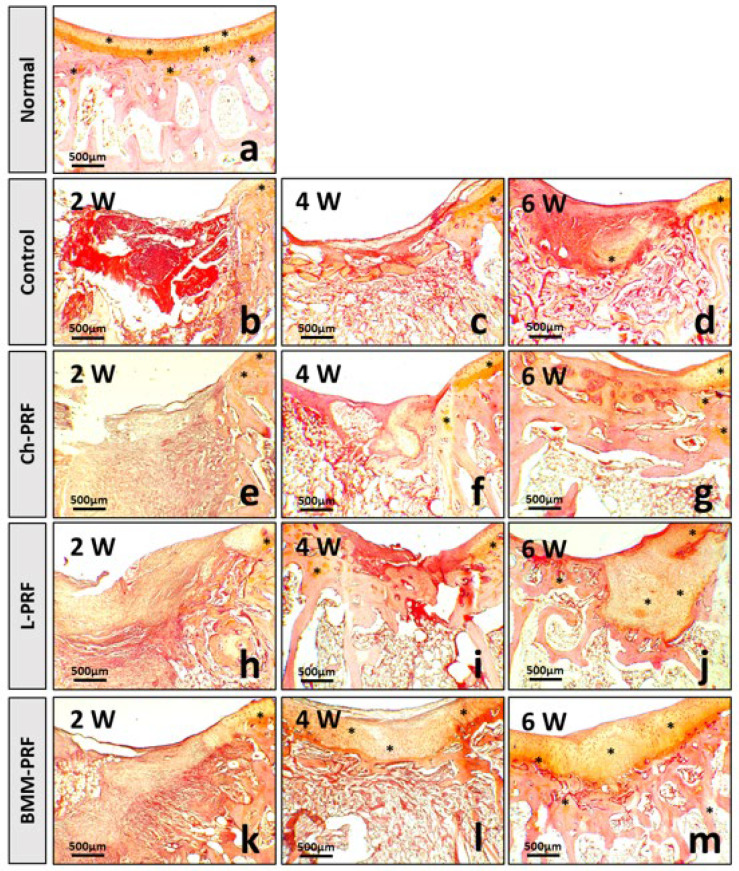
Cartilaginous proteoglycans detection by safranin O staining. Normal cartilage tissue from the knee joints of rabbits (**a**). Cartilage regeneration at 2, 4, and 6 weeks postoperatively in the control (**b**–**d**), Ch-PRF (**e**–**g**), L-PRF (**h**–**j**), and BMM-PRF (**k**–**m**) groups. Safranin O binds to cartilaginous proteoglycans and shows an orange-yellow color. The asterisk indicates cartilaginous proteoglycan-positive staining.

**Figure 10 bioengineering-10-00661-f010:**
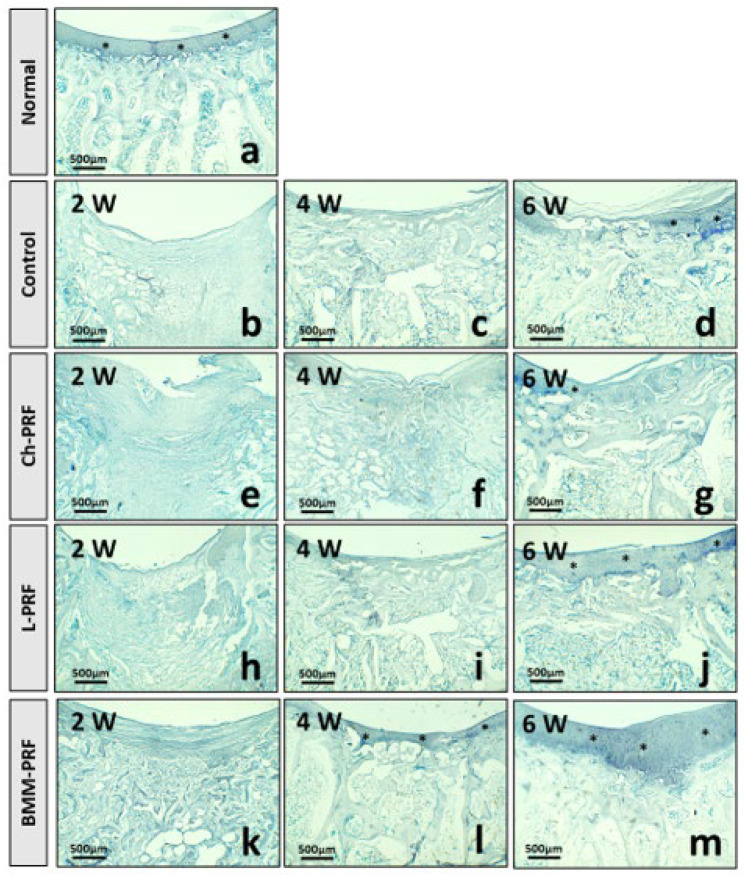
Cartilaginous proteoglycans detection by toluidine blue staining. Normal cartilage tissue from the knee joints of rabbits (**a**). Cartilage regeneration at 2, 4, and 6 weeks postoperatively in the control (**b**–**d**), Ch-PRF (**e**–**g**), L-PRF (**h**–**j**), and BMM-PRF (**k**–**m**) groups. Toluidine blue binds to cartilaginous proteoglycans and shows a blue color. The asterisk indicates cartilaginous proteoglycan-positive staining.

**Figure 11 bioengineering-10-00661-f011:**
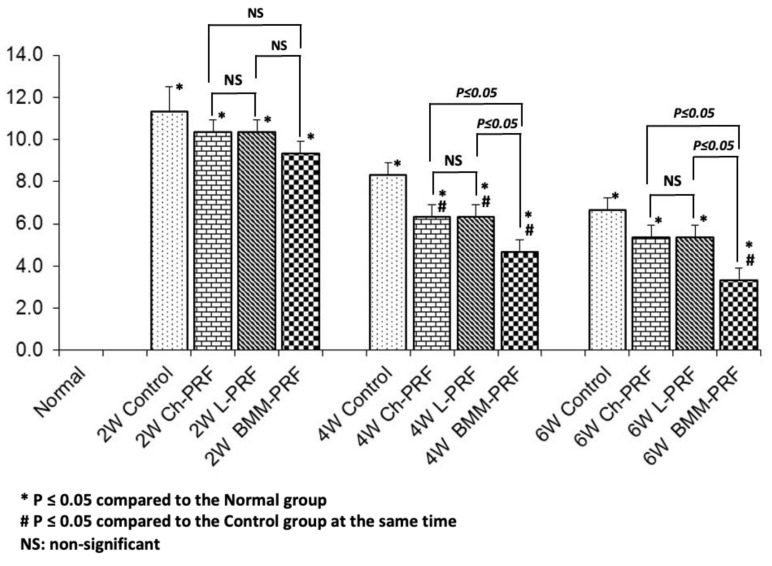
Evaluation of histological regeneration of osteochondral defects at 2, 4, and 6 weeks following surgery using the Wakitani scale. (* *p* ≤ 0.05 compared to Normal group; # *p* ≤ 0.05 compared to Control group at the same time).

**Table 1 bioengineering-10-00661-t001:** Primer sequences.

Gene	Primer Name	Forward	Reverse
18S-rRNA	r18S	ATCAGATACCGTCGTAGTTC	TTCCGTCAATTCCTTTAAG
SRY-box transcription factor 9	rSOX9	GCTCCGACACCGAGAATACA	TTGACGTGGGGCTTGTTCTT
Alkaline Phosphatase	rALPL	ACTGTGGACTACCTCTTG	GGTCAGTGATGTTGTTCC
Aggrecan	rACAN	TGGAGAAGCCCTTGCATCTG	TGGGACGGAGGATGCTTCTA
Bone Gamma-Carboxyglutamate Protein	rBGLAP	ACTCTTGTCGCCCTGCTG	CTGCCCTCCCTCTTGGAC
RUNX family transcription factor 2	rRUNX2	TCAGGCATGTCCCTCGGTAT	TGGCAGGTAGGTATGGTAGTGG
Collagen type I alpha 1 chain	rCOL1A1	GAGGTGGACACCACCCTCAA	CCAGTGTCCATGTCGCAGAA
Collagen type II alpha 1 chain	rCOL2A1	CTGTCCTGTGCGACGACATA	TCCTTTCTGCCCCTTTGGTC

## Data Availability

Not applicable.

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
