# Peer review of "Osteochondral Regeneration Ability of Uncultured Bone Marrow Mononuclear Cells and Platelet-Rich Fibrin Scaffold"

_bioengineering, 2023, doi:10.3390/bioengineering10060661_

Round 1

Reviewer 1 Report

Authors have conducted this in vivo study to investigate the effectiveness of PRF scaffolds and autologous uncultured bone marrow mononuclear cells on osteochondral regeneration in rabbit knees. 

Abstract:

All study groups and the 3 different PRF modifications are clearly stated. The aim of the study is clear. The interventions and the evaluation periods are clearly reported. Their abstract is comprehensive and well-structured.

Keywords:

I suggest authors add the following keywords to their list: “tissue regeneration” and “osteogenesis”.

Introduction:

Lines 46 and 47: 

“Cartilage regeneration remains a challenge in tissue engineering because it does not contain neural and vascular 46 components and thus displays limited restoration after injury [1]” 

For a statement as comprehensive as this, authors need to refer to the newest published studies. A reference from 2012 is simply not acceptable. This statement with this dated reference, as it is, only means that cartilage regeneration was still a challenge in 2012. Authors need to refer to 2023 published studies to back their statements. I suggest authors search for 2022/2023 reviews on this subject to show that regardless of the numerous in vivo and clinical studies executed on inducing osteochondral regeneration, it is still a very complex challenge with a lot of unanswered questions.

Lines 68, 69, 70, and 71:

“Chondrogenesis depends on several cartilage-specific markers, such as collagen 68 type I and II (COL1A1 and COL2A1), aggrecan (ACAN), and SRY-box transcription factor 9 (SOX9), and osteogenic- 69 related markers, such as Alkaline Phosphatase (ALPL), Bone Gamma-Carboxyglutamate Protein (BGLAP), and RUNX 70 family transcription factor 2 (RUNX2), which participate in tissue regeneration.”

These are not personal opinions of authors, or their reported outcomes. These are assumingly facts and statements that authors have learned through other sources and studies. Hence, authors definitely need to refer to newly-published studies that have indicated and reported these “facts”.

Overall, I believe for a subject as significant and new as tissue regeneration and osteogenesis, the references of the introduction of this study can be significantly improved. There are only 18 references in the introduction which can be a lot more given the diversity and depth of this subject. Additionally, a lot of the references are published before 2020, I suggest authors add a lot of newly-published studies in their introduction. Doing so, their judgements on the current status of their investigated subject and their rational for this study would be significantly more justifiable. Any statement that is not directly reported as authors’ results or opinion, has to be properly referred to newly-published studies. In addition, I believe this introduction is a bit short compared to the large scale of in vitro, in vivo and clinical studies that have been executed for osteochondral regeneration. Major revisions are required for this introduction, regarding its length and replacing its dated references with new studies.

Materials and Methods:

Authors have done a decent job in indicating all of their methods and materials.

Results:

Authors have properly reported their outcomes section by section. There was not a single part of their investigation that was indicated in their methods, that did not get fully reported in their results. All of the key findings are reported. Their figures and charts are appropriately placed. The quality of their histopathological graphical findings is very good.

Discussion:

Same problem as introduction, references are very limited and dated. As long as authors do not include the newest published studies, their judgements, comparisons and suggestions are not of much value. 

Overall, I believe this study has a lot of potential. Their investigations are valuable and their results are promising for future advancements in osteochondral regenerations in vivo and in human clinical studies. Authors have carefully designed their methods; their population and interventions are clearly reported. Their results are comprehensive and does not fail to report key findings. However, the references utilized for their introduction and discussion are mostly dated. There are a lot of statements and comparisons made in their introduction and discussion which do not have any references. The poor referencing and lack of newly-reported evidence in their introduction and discussion, significantly undermines their valuable investigations and presented outcomes. I strongly suggest authors apply major changes to their introduction and discussion and upgrade most of their references. I suggest accepting this paper after major revisions. Their paper would only be acceptable if they comply to all of the suggested changes. 

Author Response

Reviewer 1:

Abstract:

All study groups and the 3 different PRF modifications are clearly stated. The aim of the study is clear. The interventions and the evaluation periods are clearly reported. Their abstract is comprehensive and well-structured.

 Response: Thank you very much for your time, consideration as well as your useful comments.

Keywords:

I suggest authors add the following keywords to their list: “tissue regeneration” and “osteogenesis”.

 Response: Thank you for your comment! We have revised the keywords as per your comment

Introduction:

Lines 46 and 47:

“Cartilage regeneration remains a challenge in tissue engineering because it does not contain neural and vascular 46 components and thus displays limited restoration after injury [1]”

For a statement as comprehensive as this, authors need to refer to the newest published studies. A reference from 2012 is simply not acceptable. This statement with this dated reference, as it is, only means that cartilage regeneration was still a challenge in 2012. Authors need to refer to 2023 published studies to back their statements. I suggest authors search for 2022/2023 reviews on this subject to show that regardless of the numerous in vivo and clinical studies executed on inducing osteochondral regeneration, it is still a very complex challenge with a lot of unanswered questions.

 Response: Thank you for your comment! We have added two recent references (Aging Dis 2021 and Acta Biomateriali 2022)

Lines 68, 69, 70, and 71:

“Chondrogenesis depends on several cartilage-specific markers, such as collagen 68 type I and II (COL1A1 and COL2A1), aggrecan (ACAN), and SRY-box transcription factor 9 (SOX9), and osteogenic- 69 related markers, such as Alkaline Phosphatase (ALPL), Bone Gamma-Carboxyglutamate Protein (BGLAP), and RUNX 70 family transcription factor 2 (RUNX2), which participate in tissue regeneration.”

These are not personal opinions of authors, or their reported outcomes. These are assumingly facts and statements that authors have learned through other sources and studies. Hence, authors definitely need to refer to newly-published studies that have indicated and reported these “facts”.

Overall, I believe for a subject as significant and new as tissue regeneration and osteogenesis, the references of the introduction of this study can be significantly improved. There are only 18 references in the introduction which can be a lot more given the diversity and depth of this subject. Additionally, a lot of the references are published before 2020, I suggest authors add a lot of newly-published studies in their introduction. Doing so, their judgements on the current status of their investigated subject and their rational for this study would be significantly more justifiable. Any statement that is not directly reported as authors’ results or opinion, has to be properly referred to newly-published studies. In addition, I believe this introduction is a bit short compared to the large scale of in vitro, in vivo and clinical studies that have been executed for osteochondral regeneration. Major revisions are required for this introduction, regarding its length and replacing its dated references with new studies.

 Response: Thank you for your comment! We have added several references indicating the role of collagen, aggrecan, SOX9, ALPL, BGLAP, RUNX2 in Chondrogenesis

Materials and Methods:

Authors have done a decent job in indicating all of their methods and materials.

 Response: Thank you very much for your time, consideration as well as your useful comments.

Results:

Authors have properly reported their outcomes section by section. There was not a single part of their investigation that was indicated in their methods, that did not get fully reported in their results. All of the key findings are reported. Their figures and charts are appropriately placed. The quality of their histopathological graphical findings is very good.

 Response: Thank you very much for your time, consideration as well as your useful comments.

Discussion:

Same problem as introduction, references are very limited and dated. As long as authors do not include the newest published studies, their judgements, comparisons and suggestions are not of much value.

 Response: Thank you for your comment! We have revised and cited morre references

Overall, I believe this study has a lot of potential. Their investigations are valuable and their results are promising for future advancements in osteochondral regenerations in vivo and in human clinical studies. Authors have carefully designed their methods; their population and interventions are clearly reported. Their results are comprehensive and does not fail to report key findings. However, the references utilized for their introduction and discussion are mostly dated. There are a lot of statements and comparisons made in their introduction and discussion which do not have any references. The poor referencing and lack of newly-reported evidence in their introduction and discussion, significantly undermines their valuable investigations and presented outcomes. I strongly suggest authors apply major changes to their introduction and discussion and upgrade most of their references. I suggest accepting this paper after major revisions. Their paper would only be acceptable if they comply to all of the suggested changes.

Response: Thank you very much for your time, consideration as well as your useful comments. We have revised our  as per your comments

Reviewer 2 Report

There are certain problems with this research and it is recommended to make modifications before accept it. The specific comments are as follows:(1)In line 68-69, the author states that cartilage generation relies on collagen type I , which is incorrect. Cartilage generation relies on collagen type II, but does not rely on collagen type I.(2)In line 78, please provide the specific name of the ethics committee.(3) During animal modeling, there were only 2 untreated rabbits in the normal control group, and the experiment tested 3 time points. The normal control group should have at least 1 rabbit as the normal control at each time point; Moreover, in subsequent experiments, the gross specimen, X-ray examination, and HE staining all lacked images and data of the normal control. Special staining only showed a picture of a normal specimen, and the relevant data of the normal group of rabbit specimens should be plotted according to the time point. To reduce the number of sacrificed rabbits, the contralateral limb should also be used for control. It is recommended to supplement these data and pictures.(4)For rabbits, is the 3mm * 3mm bone cartilage defect too small? Please provide an explanation and provide references.(5)Why did animal experiments choose 2, 4, and 6W as observation time points? For rabbits, these three time points are all too short. Please provide an explanation.(6) X-ray alone cannot effectively reflect the repair of bone and cartilage. It is recommended to supplement microCT related data to reflect the repair of bone.(7)It is recommended to supplement the lack of cytotoxicity tests on materials and safety experiments in animals.(8)The discussion was too simplistic and lacked in-depth explanation of possible mechanisms of the repair of cartilage and bone with the scaffold.(9)Please confirm if there is a need to modify the format of the cited references. Two references are cited as [X, Y], and multiple references are cited as [X-Z].

Only minor errors need to be corrected.

Author Response

There are certain problems with this research and it is recommended to make modifications before accept it. The specific comments are as follows:

(1) In line 68-69, the author states that cartilage generation relies on collagen type I , which is incorrect. Cartilage generation relies on collagen type II, but does not rely on collagen type I.

Response: Thank you for pointing out a mistake. We have revised it

(2) In line 78, please provide the specific name of the ethics committee.

Response: Thank you for your comment, we have added the specific name of the ethics committee, The Hue University Animal Ethics Committee

(3) During animal modeling, there were only 2 untreated rabbits in the normal control group, and the experiment tested 3 time points. The normal control group should have at least 1 rabbit as the normal control at each time point; Moreover, in subsequent experiments, the gross specimen, X-ray examination, and HE staining all lacked images and data of the normal control. Special staining only showed a picture of a normal specimen, and the relevant data of the normal group of rabbit specimens should be plotted according to the time point. To reduce the number of sacrificed rabbits, the contralateral limb should also be used for control. It is recommended to supplement these data and pictures.

Response: Thank you for your comment! The normal control group has the normal structure of cartilage as we can observe in Figure 9 and Figure 10. The cartilage structure of the normal control group does not change during the experiment time (2, 4, and 8 weeks). The ICRS scale-based macroscopic evaluation of the normal control group is 12 (Figure 4). The Wakitani scale evaluation of the normal control group is 0 (Figure 11). In Figure 3, 5-8, we would like to compare the osteochondral regeneration between the experimental group. Each Figure contains many small pictures (12 pictures from a to l), so we did not put the normal control group or contralateral limb picture. 

(4) For rabbits, is the 3mm * 3mm bone cartilage defect too small? Please provide an explanation and provide references.

Response: Thank you for your comment! The rabbit's knee is small. We have made the defect size (3mm * 3mm) similar to those described previously [Cells. 2021 Dec; 10(12): 3536. (doi: 10.3390/cells10123536)]. We have revised the Method and added the references

(5) Why did animal experiments choose 2, 4, and 6W as observation time points? For rabbits, these three time points are all too short. Please provide an explanation.

Response: Thank you for your comment! We have optimized the experimental time points as our previous publication (ACTA VET. BRNO 2022, 91: 293–301; https://doi.org/10.2754/avb202291030293). We can observe the osteochondral regeneration in the rabbit knee at 2, 4, and 6W after surgical

(6) X-ray alone cannot effectively reflect the repair of bone and cartilage. It is recommended to supplement microCT related data to reflect the repair of bone.

Response: Thank you for your comment! Microfocus computed tomography (micro CT) is a valuable method for measuring osteochondral regeneration. Our research limitation is not provided micro CT data.  We provide the osteochondral regeneration data through real-time PCR,  xray, and histological staining (HE, Safranin O, and Toluidine Blue staining) as well as using ICRS scale-based and the Wakitani scale system.

(7) It is recommended to supplement the lack of cytotoxicity tests on materials and safety experiments in animals.

Response: Thank you for your comment! Cytotoxicity tests on materials and safety experiments in animals are necessary tests for biomaterial experiments. In our experiment, we use autologous uncultured bone marrow mononuclear cells and an autologous platelet-rich fibrin scaffold. The autologous cell and autologous bioscaffold were directly transplanted into their own rabbit. We did not use the exogenous material. in this study, the autologous cells and autologous bioscaffold may not have cytotoxicity and it is maybe safe for their own animal

(8) The discussion was too simplistic and lacked in-depth explanation of possible mechanisms of the repair of cartilage and bone with the scaffold.

Response: Thank you for your comment! We have revised the Discussion with an explanation of the potential of PRF's therapeutic effect, and bone marrow mononuclear cell therapy, which may implicate our positive result about osteochondral regeneration using bone marrow mononuclear cell and PRF scaffold.

(9) Please confirm if there is a need to modify the format of the cited references. Two references are cited as [X, Y], and multiple references are cited as [X-Z].

Response: Thank you for your comment! Could we revise it in the final blank text?

Round 2

Reviewer 2 Report

I have no other comments.

Minor editing of English language required.

Author Response

Thank you very much for your time, consideration, and useful comments!